# OpenReview forum: "The Minimal Search Space for Conditional Causal Bandits"
_ICLR.cc/2026/Conference — Submitted to ICLR 2026_

### Official Review · Reviewer_NYL5 · 2025-10-28

**Soundness:** 2
**Presentation:** 3
**Contribution:** 2
**Rating:** 6
**Confidence:** 2

**Summary:**

This paper studied the problem of finding a minimal action space in the contextual causal bandit problem with single-node interventions. An efficient algorithm is proposed to identify the minimal globally interventionally superior set. Experimental results characterize its empirical benefits over brutal force methods.

**Strengths:**

- The paper investigates the minimal action space in causal bandits under contextual decision-making. The problem is novel and of potential interest to the community.
- The presentation is clear with intuitive examples.

**Weaknesses:**

- The problem requires single-node intervention without unobserved confounders, which limits the applications. But it still has theoretical contributions to the study of contextual causal bandits.
- Some concerns remain regarding the correctness of the theoretical results, and they are discussed in Questions.

**Questions:**

- Proposition 4 seems to be a core theoretical result. I could misunderstand some points, but it is unclear why the conditional-intervention superiority partial order can be independent of the choice of observable conditioning set. In the example of Figure 1(a), if $A_2 \notin \boldsymbol{Z}_{A_1}$ but $A_2$ can be observed when intervening $X_3$, $X_3$ might be a better node. Are there any constraints on the observable conditioning set?

- I would suggest moving the related works after the problem is formulated. A detailed comparison with Lee & Bareinboim (2020) should be given. The limitation of their method and why it can not solve the contextual bandit problem is not clear. If scopes can be compared, why can't the comparison be used to search for a smaller space containing the optimal intervention?

- Are all the nodes other than $Y$ intervinionable? If some are not, mGISS could be different.

- $V$ should be $W$ in Definition 1.

---

> ### Author Response · Authors · 2025-11-19
>
> Thank you for your insightful review and questions. We address the weaknesses and questions below.
>
> ## Answers to Weaknesses
> 1. Indeed, we chose to tackle the problem of single-node interventions. We see our paper as a necessary step towards the more general case of a finite number $K$ of interventions (see also our answer to question 2. of the reviewer Pzei). Furthermore, our method indeed requires the causal graph to be known, which will not hold in many applications. However, identifying the minimal search space in this setting is itself a significant and nontrivial contribution. Extending the framework to settings with hidden confounders is a natural next step, but even the latent confounder-free case is complex, and addressing latent structure would demand substantial additional work (towards which this paper is a first step) beyond this paper's scope.
> 2. See the answers to the Questions below.
>
> ## Answers to Questions
> 1. There is indeed a constraint on the observable conditioning set that, although alluded to in the traffic controller and kidney function examples (see paragraph on conditional interventions in Section 2), was used in the proof of Proposition 4, should be explicitly/mathematically stated in the Preliminaries. We will add that to the paper. The assumption is that the context that is available for an intervention is also available for later interventions. In the traffic controller case, all the information that was available when intervening on the delay $D_{i,s}$ of a train $i$ at station $s$ is also available when intervening on the delay $D_{i,s+1}$ at the next station. In the kidney function example, all the data available when performing an intervention on the node RenalFlow1 of week 1 will also be available when intervening on the node RenalFlow2 of week2. Mathematically, this is to say that $W \in \mathrm{An}(X) \Rightarrow Z_{W} \subseteq Z_{X}$. Thank you for pointing out to us how this constraint on the observable conditioning sets was not clear, and would indeed lead to anomalies such as those described in the example that you provided.
> 2. The approach of Lee & Bareinboim (2020) is not applicable to our problem in two ways. First, it is "too ambitious": by not restricting the conditioning sets (nor the sets of intervenable nodes) in any way, a graphical characterization of the search space is not obtained. We instead focus on the case (realistic in many applications, but not realistic in others) where one knows that the conditioning sets are of the form discussed in the preliminaries. This allows us to arrive at a minimal search space by sacrificing some generality. Second, their work is for interventions on an unlimited amount of 1-node interventions, not for a budget of a single-node intervention.
>     We have attempted to explain this concisely in the related work section, but we agree that this can be further expanded upon in the paper.
>     Finally, please note additionally that we are not concerned with finding the smallest conditioning sets we can - as described in section 2 of the paper, these are fixed by the experiment/intervener. That said, in practice your observation is correct that one may want to speed up the bandits algorithm that is downstream from our method by using the results from Lee & Bareinboim (2020) to reduce the sizes of some of the conditioning sets.
> 3. Yes, exactly. The case with non-intervenable nodes is a different problem, for which this paper is a necessary step, in much the same way that [1] is a first step towards [2].
> 4. We think you are referring to the $V$ in $Z_V$? If so, yes, we should have written for $Z_W$ (and $Z_X$) instead of $Z_V$. We will correct that typo.
>
> [1] 2018 Lee, Sanghack and Bareinboim, Elias Structural causal bandits: Where to intervene?
>
> [2] 2019 Lee, Sanghack and Bareinboim, Elias Structural causal bandits with non-manipulable variables

---

### Official Review · Reviewer_Pzei · 2025-10-29

**Soundness:** 4
**Presentation:** 3
**Contribution:** 3
**Rating:** 6
**Confidence:** 3

**Summary:**

The paper extends existing causal bandit theory from hard interventions to conditional interventions, where the value of an intervened variable can depend on observed contextual variables. This formulation better captures many real-world decision-making problems under causal uncertainty. The authors formally define the conditional causal bandit problem, in which an agent knows the causal graph but not the underlying structural functions or noise distributions. To study this setting, the paper introduces a graphical characterization of the minimal set of nodes guaranteed to contain the optimal conditional intervention—those most worth exploring to maximize the expected reward. The authors further propose an efficient linear-time algorithm (C4) that identifies this minimal search space directly from the causal graph. Empirically, C4 significantly reduces the number of nodes that must be tested, lowering computational costs and accelerating convergence when integrated into standard multi-armed bandit algorithms.

**Strengths:**

Please find the strengths below:
1. The paper first generalizes causal bandit theory from hard interventions to conditional interventions, which are more realistic for some real-world decision-making scenarios involving contextual information. The paper provides a formal and precise definition of the conditional causal bandit problem.
2. The paper provides a graphical characterization of the minimal search space and provides C4 algorithm to find it with linear time complexity.
3. Comprehensive experiments on both synthetic and real-world causal graphs for C4 algorithm are demonstrated.

**Weaknesses:**

Please find the weaknesses below:
1. The framework assumes that the causal graph is fully known to the agent, which is often unrealistic in a lot of practical settings where causal structure must be learned or inferred. In contrast, several recent studies have relaxed this assumption and investigated causal bandit settings without requiring a known graph or even its skeleton, such as

Feng, Shi, Nuoya Xiong, and Wei Chen. "Combinatorial Causal Bandits without Graph Skeleton." The 16th Asian Conference on Machine Learning (Conference Track).

Yan, Zirui, and Ali Tajer. "Linear causal bandits: Unknown graph and soft interventions." Advances in Neural Information Processing Systems 37 (2024): 23939-23987.

2. The study focuses only on single-node conditional interventions, while some related works, including the papers mentioned above, have explored multi-node or joint interventions that are more general and better suited to complex causal decision-making settings.
3. The paper lacks comparisons with general contextual or combinatorial bandit baselines and does not analyze how graph properties (e.g., sparsity or depth) influence C4’s pruning ratio or regret improvement.

**Questions:**

The questions are related to the weaknesses:
1. Can the framework be extended to cases where the causal graph is unknown or partially known?
2. Can the method be generalized to multi-node or joint interventions?
3. What kinds of graphs make C4 more effective in pruning and regret reduction?

---

> ### Author Response · Authors · 2025-11-19
>
> Thank you for the thoughtful review and comments. We address the stated weaknesses and asked questions below.
>
> ## Answers to Weaknesses
>
> 1. Indeed, our method requires the causal graph to be known, which may not hold in all applications.
>    We see our work as complementary to causal discovery methods, and focuses on a different problem.
>    Causal discovery methods can be applied to identify the causal graph after which our approach can be used to find the minimal search space.
>     However, establishing the minimal search space in this setting is already a significant and nontrivial contribution.
>     Regarding the papers by Feng et al. and Yan et al., these address causal bandit algorithms with hard interventions on multiple nodes. Our paper studies the *algorithms for finding the minimal search space* for conditional interventions on single nodes, so the settings are not directly comparable. As we will argue below and have mentioned in the paper, the single node setting is actually more complex than the multi-node setting. Moreover, those works develop causal bandit algorithms, whereas our contribution is a structural reduction that can serve as a pre-processing step for any intervention-selection algorithm, again making their approaches not comparable with ours.
> 2. Please note that, perhaps surprisingly, the case of causal bandits with multi-node interventions are actually *not* more general than the case of single-node interventions. One problem is not subsumed by the other. For an example, see the last paragraph of our comment for reviewer g5Fp. In that case, if one is interested in multi-node interventions, the minimal search space consists of a single set of nodes $\lbrace A_1, A_2 \rbrace$ (the parents of the target node $Y$). If one instead is interested in single-node (conditional) interventions, the minimal search space is mGISS$=\lbrace A_1, A_2, X\rbrace$. Furthermore, finding a minimal search space for the single-node case is, again surprisingly, actually much harder than finding the minimal search space for the multi-node case (see the last paragraph of Section 2, and also the last paragraph of our comment for reviewer g5Fp). Finally, in many applications one cannot intervene on as many nodes as one wants, and is instead restricted to a single intervention, or a few interventions. If one is restricted to a single intervention, our framework can be used. The case where there's a finite number (but larger than 1) of interventions that can be performed is a generalization of which case remains whose minimal search space remains to be explored, and towards which our paper is a solid first step.
> 3. Please note that, to our knowledge, no algorithms address the conditional causal bandits problem, making any comparison impossible. More importantly, such a comparison would not be appropriate, because our method is not a bandit algorithm but a pre-processing step that identifies the minimal set of nodes that could contain the optimal conditional intervention. Therefore, we cannot compare it with (conditional causal) bandit algorithms, since it is not one. The purpose of the experiment was instead to show the effect of applying our search-space reduction as a pre-processing step to a (contextual) UCB bandit algorithm. The positive results indicate that this reduction is meaningful and can improve causal bandit algorithms. This is the same reason why [1] (which studies an analogous problem for atomic causal bandits) compares only with a brute-force or trivial algorithm, even though atomic causal bandit algorithms already existed at the time. In summary, comparing (mGISS + UCB) with an existing causal bandit algorithm would be inappropriate and, in fact, impossible because no such algorithm exists for conditional interventions. While it could be appropriate or interesting to examine the impact of using mGISS as the search space for a conditional causal bandit algorithm, that algorithm does not yet exist. About how graph properties influence C4's impact: see our answer to your question 3. below.
>
> [1] 2018 Lee, Sanghack and Bareinboim, Elias Structural causal bandits: Where to intervene?

---

> > ### Author Response · Authors · 2025-11-19
> >
> > ## Answers to Questions
> > 1. As mentioned above, our work focuses on the problem where one uses causal information to find a minimal search space, so there is no clear way to extend this framework to cases where the causal graph is unknown. In cases where it is partially known, a simple adaptation would be to run the C4 algorithm on each causal graph consistent with the known part of the graph, and take the union of the resulting minimal search spaces.
> > 2. We expect that a similar approach would be possible for the case with a finite number $K$ of interventions, but not in a trivial way: for instance, we do not expect the minimal search space to simply be the cartesian product of the LSCA closures, as one could at first suspect. For example, for the 2-node intervention setting, if two nodes $A_1, A_2 \in Pa(Y)$ are related only by a confounder $C$ (such that $A_1 \leftarrow C \rightarrow A_2$), then $(C, A_1)$ and $(C, A_2)$ are in the cartesian product $\mathrm{mGISS} \times \mathrm{mGISS}$, but clearly $(A_1, A_2)$ is superior to either of those pairs of nodes. We see our paper as a necessary stepping stone for that problem.
> > 3. We performed experiments which indicate that lower edge density tend to lead to more effective pruning, and this was the case in realistic graphs (from the bnlearn repository). See Figures 5 and 6, and the discussion in Section 6. As noted by the reviewer, we indeed did not test how sparsity and depth related to C4's pruning ratio, but we did test the impact of edge density, node number and average degree, which seemed to us to be the relevant graph properties in this case. Please let us know if and why you still think that depth should be seen as a relevant parameter to test. Finally, please note that the relation with regret improvement highly depends on the structural functions (which are unknown - we only have the causal graph). It is thus highly unclear how to test it, but we of course welcome suggestions from the reviewer if they have some.

---

### Official Review · Reviewer_g5Fp · 2025-11-01

**Soundness:** 2
**Presentation:** 3
**Contribution:** 2
**Rating:** 4
**Confidence:** 4

**Summary:**

This paper studies the conditional intervention selection problem with no latent confounders, and single-variable interventions when the causal graph is provided. The goal is to use the graphical structure to identify the variables where no conditional intervention could be optimal, thereby  reducing the search space of the ensuing bandit problem. For each variable X, they assume some fixed conditioning set that is chosen by the practitioner and adhere to the constraints of the graph (in included it X’s ancestors, does not include X’s descendants).

The main contribution is defining the notion of an optimal minimal intervention set in the conditional intervention context then deriving a graphical algorithm to find it.

**Strengths:**

One of the most important techniques in the causal bandit literature has been the minimal intervention set. Generalizing this notion to continuous interventions, the authors make an important contribution.

**Weaknesses:**

For the formulation: I think that by allowing g to be arbitrary, most of the interesting structure of the problem is lost. Roughly, where is the tension between intervening on ancestors and descendents?

For the experiments: only the trivial baseline is included. This baseline ignores the structure of the problem, so of course the proposed algorithm does better. You need to include other baselines that, at the least, try naive approaches to reduce the search space. Since no analysis of UCB is provided, only the empirical results demonstrate the efficacy of the method. Would you instead argue that the graphical contributions stand on their own?

**Questions:**

Definition 1: what is the quantification over conditioning sets? There is a “For every Z_V” in the statement but no Z_V in (1). Do you mean for every Z_X, Z_W?
Definition 2 does not depend on the conditioning set. By showing the two equivalent, does this mean that the conditioning sets don’t matter? I find it hard to believe that there are not examples where the conditioning set does not change which intervention is superior.

Why is the equivalence of definition 1 and 2 surprising? There is no restriction on the function class g, so we could set g to be the pointwise maximum as a function of the conditioning set.

Can you give some intuition for how mGISS criteria differ from the mMIS criteria? For what graphs are they equivalent? Can you give a compelling example when they are different and when the difference is necessary to learn properly?

---

> ### Author Response · Authors · 2025-11-17
>
> Thank you for the detailed assessment and constructive questions. We are glad you find our contributions important. We address your concerns below. Because of the word limit per comment, and to not sacrifice the clarity and comprehensiveness of our answer, we will divide our answer in two comments.
>
> ## About the formulation:
> Regarding your comment: "I think that by allowing g to be arbitrary, most of the interesting structure of the problem is lost. Roughly, where is the tension between intervening on ancestors and descendents?": if the concern is that it is always better to intervene on an ancestor than a descendant or vice versa, the solution space is actually significantly more complex than that. For example, consider a causal graph with nodes $A, A', D, D', C, C', Y$, and edges $A \rightarrow D \rightarrow C \rightarrow Y$, $A' \rightarrow D' \rightarrow C' \rightarrow Y$, $A\rightarrow D'$ and $D'\rightarrow C$. Then $A$ is an ancestor of $D$ and $A'$ is an ancestor of $D'$, but $A$ and $D'$ are in the minimal search space, while $A'$ and $D$ are not.
> If the concern is that conditional interventions are too general, we understand why it may seem that way, since we indeed do not make any assumptions about $g$. But this actually means that the minimal search space remains valid regardless of how a downstream bandit algorithm (or other type of intervention selection algorithm) parameterizes or learns $g$, which is a desirable property of our framework. If we have misunderstood the meaning of your comment, we ask if you could please clarify and we'll be happy to address the concern.
>
>
>
> ## About the experiments and baselines:
> As far as we know, there are no algorithms that tackle the conditional causal bandits problem, so that a comparison would not be possible.
> More importantly, such a comparison would actually not be appropriate, because our algorithm is not a bandit algorithm, but a pre-processing step that characterizes the minimal set of nodes that may ever contain the optimal conditional intervention.
> Thus, we could not compare with (conditional causal) bandit algorithms, since ours is not one.
> Instead, the goal of the experiment was to illustrate the impact of using our search space reduction as pre-processing step for a (contextual) UCB bandit algorithm.
> The fact that our results were positive provides evidence that this reduction of search space is meaningful and can improve causal bandit algorithms.
> This is the same reason why in [1] (which tackles a similar problem but for atomic causal bandits) they also only compare with a brute-force/trivial algorithm, even though there existed atomic causal bandit algorithms at the time of writing of that article.
> So, to conclude, comparing (mGISS + UCB) with an existing causal bandit would be inappropriate, and actually not possible since such algorithm does not exist for conditional interventions.
> It could be appropriate/interesting to see the impact of using the mGISS as search space for a conditional causal bandit algorithm, but again the latter does not exist.
> Finally, as a comment to your last question: yes, the graphical contributions stand on their own, and the minimal search space found by algorithm C4 would be valid as pre-processing step for any intervention selection algorithm for conditional interventions, contextual UCB being just one example of those.
>
> [1] 2018 Lee, Sanghack and Bareinboim, Elias Structural causal bandits: Where to intervene?

---

> ### Author Response · Authors · 2025-11-17
>
> ## Answers to questions
>
> Definition 1: indeed, we mean for every $Z_X$, $Z_W$. We agree that writing it in that way is clearer and we will rephrase accordingly.
>
> Definition 2 and the equivalence: Exactly: the conditioning sets "don't matter" (as long as they are *observable conditioning sets* in the sense described in the preliminaries. See also our answer to the first question of reviewer NYL5, about the restrictions on these observable conditioning sets). The equivalence is rigorously proved in appendix D, and it is indeed a non-trivial result, and a theoretical contribution of this paper in itself.
> We hypothesized this equivalence after studying a number of particular cases where the ancestors of $X$ were sufficient to determine a minimal search space, and confirmed our suspicions rigorously, in the form of a proof.
> Although it may at first seem to contradict the reader's intuition, we urge the reviewer to investigate our proof of the equivalence between superiority relations (Proposition 4), and check that it indeed holds.
>
> More on the equivalence: we would also add that we see this equivalence as "surprising" partly because of what you yourself observed to be non-intuitive for you: that this implies that different observable conditioning sets would result in the same superiority relations. Furthermore, it is not at all clear at first sight that conditional interventions conditioned on the observable conditioning set respect the same superiority relation as atomic interventions with knowledge of the exogenous variables - and thus it is "surprising". We do not see how setting "g to be the pointwise maximum as a function of the conditioning set" relates to the discussion, or would help making this equivalence trivial.
>
> I assume you are referring to the minimal intervention sets (MISs) from [1]? If so, the appropriate comparison would be with the POMISs (Possible Optimal Minimal Interventions Sets), not with the MISs.
> We have addressed this distinction in Section 1 and also at the end of Section 2. As further explanation: while in [1] their minimal search space would consist of the only POMIS $\mathrm{PA}(Y)$ (whenever $Y$ is not confounded by a latent confounder), our case is not as simple, even when there are no latent confounders.
> While they could simply perform a multi-node hard intervention on the whole of $\mathrm{PA}(Y)$, we (being restricted to single-node inteventions) cannot do the same - which leads us to the discussion on section 4 of our paper and the derivation of the LSCA closure.
> Furthermore, it is not meaningful to say when our mGISS would coincide with their minimal search space, because the mGISS consists of nodes, while their search space is a set of sets of nodes.
> That said, the difference becomes even more apparent if one takes a case where the causal graph has edges $X \rightarrow A_{1}$, $X \rightarrow A_{2}$, $A_{1} \rightarrow Y$ and $A_{2} \rightarrow Y$, where the minimal search space for their problem is simply $\{\{A_1, A_2\}\}$, and the minimal search space for our problem is $\{A_1,A_2, X\}$.
> For a concrete SCM with this graph, consider the case where $Y = \neg (A_{1} \vee A_{2})$, $A_{1} = \delta_{X,0}$, $A_{2} = \delta_{X,2}$, $X = N_{X} \sim \mathcal{U}[0,1,2]$ (all variables are binary except for $X$, and $\delta_{x,y} = 1 \text{ if } x=y \text{ and zero otherwise}$). Notice that $Y = 1$ iff $A_{1} = A_{2} = 0$.
> Then, conditionally intervening on $A_{1}$ would at most lead to $\mu_{Y} = 2/3$, since the case $X=2$ would lead to $A_{2}=1$, and thus $Y=0$.
> Similarly for interventions on $A_{2}$.
> However, an intervention setting $X$ to $1$ (empty observable conditioning set since $X$ has no ancestors) would lead to $\mu_{Y}=1$.
> Hence, $X$ does need to be added to the mGISS, while it is not part of the POMIS from [1].
> We can add such an explicit example to the paper.
>
> [1] 2018 Lee, Sanghack and Bareinboim, Elias Structural causal bandits: Where to intervene?

---

### Author Response · Authors · 2025-12-02
**Final Summary**

The reviewers recognized the novelty, soundness and significance of the paper, and praised the clarity of the presentation. Two of the 3 reviewers were especially
enthusiastic about the paper, and posed some minor questions asking mostly to clarify the scope of the paper. We believe that we have successfully addressed all
observations and weaknesses mentioned, and clarified all points. Thus, we are confident that these two reviewers (Pzei and NYL5) would likely increase their score.
The other reviewer (g5Fp), who is less enthusiastic about our paper, recognizes that we make an important contribution, but points out weaknesses that stem from a
misunderstanding of parts of our work and the related literature. In particular, this reviewer was very critical of the fact that we do not compare our approach to bandit
algorithms that use the causal structure (causal bandit algorithms). This misses the fact that our method is not a causal bandit algorithm, and thus such a comparison
would not be appropriate. Furthermore, a causal bandit algorithm tackling the case of conditional interventions does not exist in the current literature, so it would in
any case not be possible to do such a comparison. Reviewer Pzei, in contrast, recognized the suitability of our experiments and mentioned that “Comprehensive
experiments on both synthetic and real-world causal graphs for C4 algorithm are demonstrated.”
For details, please see the rebuttal comments we wrote to all the reviewers.

For details, please see the rebuttal comments we wrote to all the reviewers.

---

### Meta-Review · Area_Chair_2MmC · 2026-01-07

**Summary:**

The reviewers have raised concerns about scope and assumptions, such as the assumption that the causal graph is known, that there is no latent confounding, and that it focuses on single-node conditional interventions, asking whether and how it might extend to unknown/partially known graphs or multi-node/joint interventions. Furthermore, the paper discusses its contribution relative to a rather trivial brute-force baseline, and the reviewers have also raised concerns about the exposition and clarity of the assumptions.

I have read the paper carefully, and it seems that the manuscript and the reviewers have all missed all the recent contributions to causal bandits in the past 2-3 years that have addressed the problem studied in a much broader scope, more limited assumptions, and stronger results. One representive work is the following. The authors are encouraged to review the literature thoroughly, including many of the references in this paper, and several recent papers in NeurIPS 2025:

Yan, Zirui, Dennis Wei, Dmitriy A. Katz, Prasanna Sattigeri, and Ali Tajer. "Causal bandits with general causal models and interventions." In International Conference on Artificial Intelligence and Statistics, pp. 4609-4617. PMLR, 2024.

**Reviewer Concerns:**

The authors have tried to address the concerns. Nevertheless, due to missing all the existing literature, the core criticisms are not addressed and the contribution of the paper remains unclear.

**Reviewer Scores:**

I believe with a discussion period as usual and with brining up the literature that the authors and reviewers have missed, the reviewers would've been more reluctant to consider the paper for acceptance.

---

### Decision · Program_Chairs · 2026-01-26

Reject